# BUB1 Inhibition Sensitizes TNBC Cell Lines to Chemotherapy and Radiotherapy

**DOI:** 10.3390/biom14060625

**Published:** 2024-05-25

**Authors:** Sushmitha Sriramulu, Shivani Thoidingjam, Farzan Siddiqui, Stephen L. Brown, Benjamin Movsas, Eleanor Walker, Shyam Nyati

**Affiliations:** 1Department of Radiation Oncology, Henry Ford Cancer Institute, Henry Ford Health, Detroit, MI 48202, USA; 2Henry Ford Health + Michigan State University Health Sciences, Detroit, MI 48202, USA; 3Department of Radiology, Michigan State University, East Lansing, MI 48824, USA

**Keywords:** TNBC, BAY1816032, BUB1, PARP, olaparib, cisplatin, paclitaxel, radiation, chemoradiation

## Abstract

**Simple Summary:**

An inhibitor of BUB1 increases the cytotoxic ability of different classes of chemotherapy, targeted agents, and radiotherapy in triple-negative breast cancer, which is the most difficult subtype of breast cancer to treat due to its lack of expression of targetable genes. The data presented here demonstrate that the cell cycle checkpoint kinase BUB1 can serve as a novel molecular target for the treatment of triple-negative breast cancer and form the basis for the development of rational combinatorial treatment approaches.

**Abstract:**

BUB1 is overexpressed in most human solid cancers, including breast cancer. Higher BUB1 levels are associated with a poor prognosis, especially in patients with triple-negative breast cancer (TNBC). Women with TNBC often develop resistance to chemotherapy and radiotherapy, which are still the mainstay of treatment for TNBC. Our previous studies demonstrated that a BUB1 kinase inhibitor (BAY1816032) reduced tumor cell proliferation and significantly enhanced radiotherapy efficacy in TNBC. In this study, we evaluated the effectiveness of BAY1816032 with a PARP inhibitor (olaparib), platinum agent (cisplatin), and microtubule poison (paclitaxel) alone or in combination with radiotherapy using cytotoxicity and clonogenic survival assays. BUB1 inhibitors sensitized BRCA1/2 wild-type SUM159 and MDA-MB-231 cells to olaparib, cisplatin, and paclitaxel synergistically (combination index; CI < 1). BAY1816032 significantly increased the radiation sensitization of SUM159 and MDA-MB-231 by olaparib, cisplatin, or paclitaxel at non-toxic concentrations (doses well below the IC_50_ concentrations). Importantly, the small molecular inhibitor of BUB1 synergistically (CI < 1) sensitized the BRCA mutant TNBC cell line HCC1937 to olaparib. Furthermore, the BUB1 inhibitor significantly increased the radiation enhancement ratio (rER) in HCC1937 cells (rER 1.34) compared to either agent alone (BUB1i rER 1.19; PARPi rER 1.04). The data presented here are significant as they provide proof that inhibition of BUB1 kinase activity sensitizes TNBC cell lines to a PARP inhibitor and radiation, irrespective of BRCA1/2 mutation status. Due to the ability of the BUB1 inhibitor to sensitize TNBC to different classes of drugs (platinum, PARPi, microtubule depolarization inhibitors), this work strongly supports the role of BUB1 as a novel molecular target to improve chemoradiation efficacy in TNBC and provides a rationale for the clinical evaluation of BAY1816032 as a chemosensitizer and chemoradiosensitizer in TNBC.

## 1. Introduction

Triple-negative breast cancer (TNBC) is a subtype of breast cancer characterized by the absence of human epidermal growth factor receptor 2 (HER2), progesterone receptor (PR), and estrogen receptor (ER) [1]. TNBC is more difficult to cure because it lacks these receptors, which are frequently targeted for treatment [2,3]. TNBC accounts for approximately 10–15% of all breast cancer cases [4]. The recurrence risk of TNBC is higher in the first few years following diagnosis, and it tends to be more aggressive [5]. While surgery, chemotherapy, and/or radiotherapy are the mainstays of treatment, intrinsic or acquired resistance results in poor clinical outcomes [6,7,8].

Currently, frontline standard chemotherapy, composed of anthracyclines, alkylating agents, and taxanes, is commonly used to treat high-risk and locally advanced TNBC. Chemotherapeutic agents induce cell death by directly or indirectly causing DNA damage [9]. Cancer cells acquire resistance to chemotherapy by enhancing DNA damage responses. Targeting DNA repair pathways can increase the tumor sensitivity to chemotherapies in TNBC [10]. For example, about 15% of TNBC patients harbor germline mutations in BRCA1 or BRCA2, making them susceptible to targeted agents such as PARP inhibitors [11]. However, chemotherapy options for patients without these mutations are currently limited. DNA damage is a critical determinant of radiation-induced cell death [12]. Radiation induces both single-strand breaks (SSB) and double-strand breaks (DSB). The ability of cells to recognize and respond to DSB is fundamental in determining the sensitivity (or resistance) of cells to radiation [13]. Following DNA damage, cell cycle checkpoints are activated to block cell cycle progression and prevent the propagation of cells with damaged DNA [14]. DSB repair consists of two major and mechanistically distinct processes: non-homologous end-joining (NHEJ) and homologous recombination (HR). Both DNA damage repair and cell cycle checkpoints are positively regulated by several kinases, and inhibition of these kinases results in enhanced radiotherapy efficacy. TNBC has a higher expression of cell cycle-related growth-regulatory molecules [15]. The molecular drivers of therapeutic resistance are complex and include increased drug resistance due to drug efflux, chemotherapy and radiation resistance due to enhanced DNA repair, senescence escape, epigenetic changes, tumor heterogeneity, abnormal tumor microenvironment, epithelial-to-mesenchymal transition (EMT), and/or changes in cell metabolism [16].

Multiple kinases, including BUB1, positively regulate both DNA damage repair and cell cycle checkpoints. BUB1 is a G2/M cell cycle checkpoint kinase that performs several functions, including accurate chromosomal segregation during cell division [17,18,19]. BUB1 dysregulation or mutations can lead to chromosomal instability and are associated with several types of cancers, including TNBC [20,21]. BUB1 is overexpressed in most solid cancers. BUB1 transcripts are significantly higher in breast cancer cell lines and in high-grade primary breast cancer tissues compared to normal mammary epithelial cells or in normal breast tissues [22,23]. Moreover, high BUB1 expression (transcript) correlates with extremely poor outcomes in breast cancer [24,25]. Increased expression of BUB1 is associated with resistance to DNA-damaging agents (i.e., chemotherapy and radiotherapy) [26]. In previous studies, we showed that biochemical or genomic ablation of BUB1 was cytotoxic in TNBC and sensitized TNBC to radiation [27]. We also demonstrated earlier that BUB1’s kinase activity exploits TGF-β signaling to drive aggressive cancer phenotypes, including cell migration, invasion, and EMT [28]. BUB1 has been identified as a potential therapeutic target for improving the effectiveness of chemotherapy, radiotherapy, and targeted therapies [29]. Pharmacological inhibition of BUB1 sensitized cancer cells to taxanes, ATR, or PARP inhibitors and significantly reduced tumor xenografts [30]. However, a role for BUB1 in improving the efficacy of radiation or chemoradiation in TNBC was not evaluated. Therefore, the main objective of this study was to ascertain if BUB1 inhibition improved the effectiveness of chemotherapy and chemoradiation in TNBC cell lines. Given BUB1’s strong correlation to aggressiveness and different classes of drugs [23], and our observation that BUB1 inhibition sensitizes TNBC to radiation and lung cancers to chemoradiation [31,32], we rationalized that combining BUB1 inhibitors with different classes of drugs (platinum, PARPi, microtubule depolarization inhibitors) would provide strong chemoradiation sensitization in TNBC.

## 2. Materials and Methods

### 2.1. Co-Expression Studies Using the cBioPortal Database

The breast cancer dataset (METABRIC, 2509 samples) from the cBio Cancer Genomics Portal (http://cbioportal.org/; URL accessed on 15 April 2024) was used to check the correlation between mRNA expression levels (Illumina HT-12 v3 microarray, 1866 samples) of BUB1 (query gene) and BRCA1, BRCA2, PARP1, PARP2, PARP3, and TP53. cBioPortal mRNA expression data are computed by comparing the relative expression of a particular gene in a tumor sample to the gene’s expression distribution in a reference population of samples [33]. This cohort’s mRNA expression correlations were examined using Spearman’s test. Correlation plots were displayed along with the regression line and estimated significance (*p* values).

### 2.2. Chemicals

A BUB1 kinase inhibitor, BAY1816032 (Catalog No. HY-103020), was obtained from MedChemExpress (Monmouth Junction, NJ, USA). Olaparib/AZD2281 (Catalog No. CT-A2281) and paclitaxel (Catalog No. CT-0502) were obtained from Chemietek (Indianapolis, IN, USA) and cisplatin (PHR1624-200MG) was sourced from Millipore Sigma (Burlington, MA, USA).

### 2.3. Cell Lines and Cell Culture

The SUM159 cell line was initially developed by Stephen P. Ethier and obtained from Sofia Merajver (University of Michigan). The HAM’s F-12 media (Catalog No. 31765305, Thermo Fisher Scientific; Waltham, MA, USA) supplemented with 5% FBS, 10 mM HEPES, 1 μg/mL Hydrocortisone, 6 μg/mL Insulin, and 1% Penicillin-Streptomycin was used to grow SUM159 cells. DMEM media (Catalog No. 30-2002, ATCC) supplemented with 10% FBS and 1% penicillin-streptomycin was used to culture MDA-MB-231 cells, which were obtained from the American Type Culture Collection (ATCC). HCC1937 cells were also obtained from ATCC and were grown in RPMI-1640 media supplemented with 10% FBS and 1% penicillin-streptomycin. All cell lines were maintained at 37 °C in a 5% CO_2_ incubator and passaged at 70% confluence. Cell passage numbers for SUM159, MDA-MB-231, and HCC1937 ranged from P20–P25. Cells were routinely tested for Mycoplasma contamination using the MycoAlert PLUS kit (Lonza, Cat. No. LT07-705).

### 2.4. Drug Treatment and Radiation

BUB1 inhibitor (BUB1i) BAY1816032, AZD2281 (Olaparib), cisplatin, and paclitaxel were dissolved in DMSO (20 mM BUB1i, 10 mM olaparib, 20 mM cisplatin, and 10 mM paclitaxel) and stored as single-use aliquots at −80 °C. Working concentrations were prepared in serum and supplement-containing medium, and cells were treated with doses ranging from 1 nanomolar (1 nM) to 100 micromolar (100 µM). Cells were irradiated one hour after drug treatment using a CIX-3 orthovoltage unit (Xstrahl Life Sciences, Suwanee, GA, USA) with a copper filter.

### 2.5. Cell Growth and Viability

SUM159 and MDA-MB-231 were seeded at a density of 2000 cells per well and HCC1937 at a density of 4000 cells per well in 96-well plates. After 24 h, cells were treated with BAY1816032, AZD2281 (olaparib), cisplatin, and paclitaxel at various concentrations for 72 h. Cytotoxicity was assessed using the alamarBlue cell viability kit (Thermo Fisher Scientific, Cat. No. DAL1100) according to the manufacturer’s protocol. Absorbance was measured at 570 nM on a Synergy H1 Hybrid Reader (BioTek Instruments, Winooski, VT, USA). The values were compared to vehicle/mock-treated cells. A non-linear regression best-fit model was used to determine the IC_50_ values (GraphPad Prism V9). Half maximal inhibitory concentration, or IC_50_, is the half-way point at which the compound completely inhibits a biological or biochemical activity. The combination index (C.I.) is used to determine the degree of drug interaction, and it was calculated using the formula: C.I = (D)1/(Dχ)1 + (D)2/(Dχ)2. Here, (Dχ)1 and (Dχ)2 are the concentrations of each drug alone to achieve the χ% effect, while (D)1 and (D)2 are the concentrations of drugs in combination to produce the same effect. C.I > 1 implies antagonism; C.I = 1 entails additivity; and C.I < 1 indicates a synergistic effect. The significance between different groups was estimated using a one-way ANOVA (GraphPad Prism V9). The findings are presented as the mean ± standard error of the mean (SEM). All experiments were performed in triplicate and were performed at least three times. *p* < 0.05 was considered statistically significant.

### 2.6. Clonogenic Survival Assay

Cells were plated in 6-well plates at different cell densities in appropriate culture media and were maintained overnight at 37 °C. TNBC cell lines were then treated with BAY1816032 alone or in combination with AZD2281 (olaparib), cisplatin, or paclitaxel and were irradiated (0, 2, 4 Gy) after one hour. The cells were fixed and stained with methanol and crystal violet after being allowed to grow for 7–15 days or until visible colonies formed. A clone of 50 or more cells was considered a colony. The radiation enhancement ratio (rER) was estimated in MS Excel using the formula: D bar of varying inhibitor concentrations/D bar of vehicle (DMSO). The above formula indicates the radiation dose required to produce a certain level of cell killing in the absence of the inhibitor (DMSO/vehicle) divided by the radiation dose required to produce the same level of cell killing in the presence of the inhibitor. Radiation sensitization was defined as rER > 1, whilst radiation resistance or protection was defined as rER < 1. The significance between different groups was estimated using ANOVA (GraphPad Prism V9). All experiments were performed in triplicate and were conducted at least three times.

## 3. Results

### 3.1. Correlation of BUB1 mRNA Expression with BRCA1/2, PARP1/2/3, and TP53 in Breast Cancer

To determine the correlation between BUB1 and the genes associated with DNA damage response or repair pathways such as *BRCA1*, *BRCA2*, *PARP1*, *PARP2*, *PARP3*, and *TP53* [34,35], we used the cBioPortal database to perform a co-expression analysis in breast cancer datasets. Spearman’s test showed a significant positive correlation between mRNA expression of *BUB1* and *BRCA1* (0.331, *p* = 4.5 × 10^−49^), *BRCA2* (0.123, *p* = 1.10 × 10^−7^), *PARP1* (0.390, *p* = 1.12 × 10^−68^), and *PARP2* (0.371, *p* = 5.67 × 10^−62^) genes in breast cancer (Figure 1A–D). A significant negative correlation was observed for *PARP3* (−0.489, *p* = 1.61 × 10^−112^) (Figure 1E)*. TP53* showed a positive correlation (6.68 × 10^−3^, *p* = 0.773), but it was not significant (Figure 1F).

### 3.2. Antiproliferative Effects of Single Agents Olaparib, Cisplatin, and Paclitaxel in TNBC Cell Lines

Our previous studies indicated that BUB1 inhibition significantly reduced the proliferation of TNBC cells but did not affect breast epithelial cells [27]. In this study, we assessed the effects of a PARP inhibitor (olaparib/AZD2281) and chemotherapeutic agents (cisplatin and paclitaxel) on TNBC cell lines. The cytotoxicity of a single agent (IC_50_) was determined using the alamarBlue assay in SUM159 and MDA-MB-231 cells. The cells were treated for 72 h with different concentrations of drugs, and the absorbance was read. At higher concentrations of the drugs, the decrease in viability was similar for both cell lines, with MDA-MB-231 generally presenting higher IC_50_ values than SUM159 for all three drugs. The IC_50_ for olaparib varied from 19.3 μM (SUM159) to 28.3 μM (MDA-MB-231; Figure 2A,B). The single agent cisplatin IC_50_ was 1.63 μM in SUM159 and 7.14 μM in MDA-MB-231 cells (Figure 2C,D), while the single agent paclitaxel IC_50_ was less than 10 nM for both cell lines (Figure 2E,F). We estimated the single agent IC_50_ of BUB1 inhibitor BAY1816032 to be less than 4 μM in these cells [27].

### 3.3. BUB1 Inhibitor Synergistically Sensitizes TNBC to Olaparib, Cisplatin, and Paclitaxel

To assess the effects of BUB1 inhibition on PARP inhibitor sensitivity and chemosensitivity in TNBC cell lines, cell viability was assessed 72 h after exposure to olaparib (10 μM), cisplatin (1 μM), and paclitaxel (1.5 nM) in combination with BAY1816032 (1 μM). Our results demonstrated that BAY1816032 significantly increased the cytotoxicity of a PARP inhibitor in TNBC cell lines (Figure 3A,B). Consistent with this, inhibition of BUB1 also increased sensitivity to cisplatin (Figure 3C,D) and paclitaxel (Figure 3E,F) compared to cisplatin or paclitaxel alone. In both cell lines, the combination effect of BUB1i and cisplatin was synergistic, although the results were not statistically significant using a one-way ANOVA (*p* = 0.19). The combination effect of BUB1i and paclitaxel in MDA-MB-231 also showed synergism, but the results were not statistically significant (*p* = 0.21; Figure 3F). The BUB1 inhibitor demonstrated strong synergistic effects with PARPi, cisplatin, and paclitaxel with a C.I. less than 1 in both cell lines. The values that are closest to the drug’s IC_70_ concentrations were used to estimate the C.I. index. This synergism was observed at approximately 10-fold lower drug concentrations than single agents. The C.I. values in SUM159 and MDA-MB-231 for 10 μM olaparib + 1 μM BUB1i were 0.55 and 0.52, 1 μM cisplatin + 1 μM BUB1i were 0.9 and 0.63, and 1.5 nM paclitaxel + 1 μM BUB1i were 0.78 and 0.75, respectively.

### 3.4. Olaparib, Cisplatin, and Paclitaxel Differentially Radiosensitize SUM159 and MDA-MB-231 Cell Lines

We evaluated the single agent radiosensitization potential of AZD2281 (olaparib), cisplatin, and paclitaxel in SUM159 and MDA-MB-231 cells before combining them with BUB1 inhibitors. The cells were exposed to olaparib (500 nM and 2.5 μM), cisplatin (750 nM and 1 μM), or paclitaxel (1 nM and 1.5 nM) for 1 h prior to irradiation, and clonogenic cell survival assays were performed as described under methods. SUM159 cells were moderately radiosensitized at a lower olaparib concentration (rER = 1.45; 500 nM), whereas MDA-MB-231 cells showed marginal radiosensitization (rER = 1.06; Figure 4A,B). Both the cell lines demonstrated increased radiosensitization at non-toxic, increased concentrations (2.5 μM; rER = 1.71 and 1.40). Although there was moderate radiosensitization with cisplatin (rER = 1.27 and 1.13; Figure 4C,D) and paclitaxel (rER = 1.16 and 1.15; Figure 4E,F) at the doses tested, the radiosensitization by single agent olaparib was much stronger in SUM159 and MDA-MB-231.

### 3.5. BUB1 Inhibitor Enhances Radiation Sensitization by Olaparib, Cisplatin, and Paclitaxel in SUM159 and MDA-MB-231 Cell Lines

Individually, olaparib, cisplatin, and paclitaxel reduced the colony formation ability in SUM159 and MDA-MB-231 cells, indicative of radiation sensitization (Figure 5A–F, red curves). BUB1 inhibitor BAY1816032 moderately sensitized these cells to radiation (Figure 5A–F, blue curves). However, combining BUB1i with olaparib (Figure 5A,B), cisplatin (Figure 5C,D), and paclitaxel (Figure 5E,F) led to enhanced radiosensitization with an increase in the rER, although the results were not statistically significant using one-way ANOVA (*p* = 0.99; Figure 5A–F, magenta curves).

The greatest reduction in the surviving fraction with the BUB1i and olaparib combination was observed in SUM159 (rER 1.65; Figure 5A), while the most reduction with the BUB1i and cisplatin combination was in MDA-MB-231 cells (rER 1.65; Figure 5D). The BUB1 inhibitor with paclitaxel increased radiation sensitization in both cell lines (rER 1.53 in SUM159 and rER 1.67 in MDA-MB-231; Figure 5E,F). These data confirm that BUB1 inhibition potentiates the cytotoxic effects of olaparib, cisplatin, and paclitaxel with radiation in TNBC cell lines.

### 3.6. BUB1 Inhibitor BAY1816032 Sensitizes BRCA Mutant TNBC Cell Line to PARP Inhibitor and Radiation

PARP inhibitors induce synthetic lethality in BRCA1/2-mutated cancers by selectively targeting tumor cells that fail to repair DSB. Since BUB1 regulates DSB repair, we examined the potential benefits of combining BUB1i with PARPi (olaparib) in a TNBC cell line that carries BRCA mutations (HCC1937). The single agent BUB1 inhibitor BAY1816032 yielded an IC_50_ of 3.56 μM (Figure 6A), while the single agent olaparib IC_50_ was estimated to be close to/higher than 100 μM (Figure 6B). We then investigated the efficacy of BUB1i in combination with olaparib in HCC1937 cells to check if the BRCA mutant cell line can become sensitive to olaparib in the presence of BUB1i. Our data indicate that BUB1i sensitized HCC1937 cells to olaparib synergistically (CI < 1) at 5-fold lower olaparib concentrations than IC_50_, although the results were not statistically significant between BUB1i 2.5 μM and the combination group using the one-way ANOVA method (Figure 6C). Inclusion of radiation further potentiated the synergistic effects of BUB1i with olaparib, but then the results were not statistically significant between BUB1i 2.5 μM and the combination group (CI 0.83, Figure 6D). We next performed a clonogenic survival assay to reconfirm the synergistic effect of the trimodality treatment in these cells. BUB1 inhibitor BAY1816032 demonstrated radiation sensitization with 100-fold less olaparib concentration than the IC_50_ in these cells (Figure 6E), but the results were not statistically significant (*p* = 0.99).

## 4. Discussion

In this study, we assessed the efficacy of a BUB1 kinase inhibitor in combination with a PARP inhibitor (olaparib), a platinum agent (cisplatin), paclitaxel, and radiation in TNBC. We observed that BUB1 sensitized TNBC to cisplatin, PARP inhibitor olaparib, and paclitaxel and improved the radiation-mediated cytotoxicity of these agents. Interestingly, we also demonstrate that BUB1i sensitized the BRCA-mutated TNBC cell line to olaparib in combination with radiation. Overall, our results provide evidence that targeting BUB1 with a PARP inhibitor, cisplatin, or paclitaxel with radiation would be a novel approach for effectively treating TNBC.

TNBC is the most aggressive type of breast cancer, generally occurs in younger women, particularly those of African ancestry, and is difficult to cure using adjuvant therapy alone [36]. The molecular drivers of therapeutic resistance in TNBC [37,38] are complex and may include increased drug efflux, enhanced DNA repair, senescence escape, epigenetic changes, tumor heterogeneity, abnormal tumor microenvironment, or epithelial-to-mesenchymal transition. Germline mutations of the *BRCA1*, *BRCA2*, and *TP53* genes encoding important components of the DNA-damage response (DDR) are associated with a high risk of TNBC [39]. Among several PARP family members, PARP1, PARP2, and PARP3 also play vital roles in DNA damage and repair processes [40], which may contribute to the anti-tumor activity of PARP inhibitors [41]. In this study, all the TNBC predisposition genes showed a significantly positive correlation with *BUB1*, except *PARP3* and *TP53* at the mRNA level (Figure 1). PARPi therapeutic effectiveness is thought to be higher in tumors that harbor germline or somatic BRCA mutations than in BRCA wt tumors. BRCA mutations or inherent tumor sensitivity to platinum agents are interpreted as signs of deficiency in DSB repair by HR and a favorable response to PARP inhibitors. However, clinical benefit from these agents is not uniform across all BRCA-mutated or platinum-responsive patients. Contrary to this, a small number of patients with platinum-resistant or BRCA wt tumors benefit from PARPi. Therefore, identification and validation of additional reliable biomarkers will help select patients that will benefit from PARPi-based therapies in the absence of BRCA mutations or platinum sensitivity.

As TNBC accounts for about 30% of breast cancer-associated deaths with a lack of specific treatment targets [42], we envision that the identification of novel molecular targets would improve the TNBC outcome. Molecularly targeted agents can enhance chemoradiation sensitivity [13]. Given BUB1’s strong correlation to aggressiveness and different classes of drugs [23] and our observation that BUB1 inhibition sensitizes TNBC to radiation and lung cancers to chemoradiation [27,31], we rationalized that combining BUB1 inhibitors would provide strong chemoradiation sensitization in TNBC. Although the effectiveness of the BUB1 inhibitor BAY1816032 was evaluated with PARPi, cisplatin, and paclitaxel in a prior study [30], the combination with cisplatin resulted in antagonistic effects, and BUB1’s potential role in improving the efficacy of chemoradiation in TNBC was not assessed.

As PARP inhibitors, taxanes, and platinum compounds are suitable treatment options for TNBC [30,43], we first investigated the cytotoxic effects of the single agent AZD2281 (olaparib), cisplatin, or paclitaxel in TNBC cell lines (SUM159, MDA-MB-231). TNBC cells demonstrated high IC_50_ (20–40 µM) for olaparib (Figure 2), which is in line with earlier findings that TNBC cells are resistant to single agent PARPi [44]. SUM159 and MDA-MB-231 cells were modestly sensitive to cisplatin and highly sensitive to paclitaxel (Figure 2). This is consistent with a previous study, which reported that BL (basal-like) subtypes were more sensitive to cisplatin than to MSL (mesenchymal stem-like) and LAR (luminal androgen receptor) [37].

Our data demonstrate synergistic effects with BUB1 inhibitor BAY1816032 in combination with cisplatin, paclitaxel, and PARPi (olaparib) (Figure 3). Our results expand the finding by Siemeister et al. [30] on BUB1 inhibition sensitizing TNBC to paclitaxel and olaparib. However, our results are opposite to their findings with cisplatin. This could be due to the limited number of drug concentrations and combinations of BAY1816032 and cisplatin in their assays [30]. PARP inhibitors radiosensitize TNBC in preclinical models [45] and have been safe and effective in high-risk TNBC patients in clinical studies [46,47]. We tested whether BUB1 inhibitors (i.e., molecularly targeted agents) would further enhance the radiosensitization by PARPi (Figure 4). Indeed, BUB1i significantly increased radiosensitization by olaparib in SUM159 and MDA-MB-231 cells (Figure 5). Not surprisingly, BUB1i also enhanced radiosensitization with cisplatin and paclitaxel (Figure 5). These findings consolidate a role for BUB1 as a molecular target for increasing the efficacy of radiotherapy [48] and are supported by encouraging results with WEE1 inhibitors that have been combined with radiation in clinical trials with promising results [49,50].

Since PARP inhibitors have recently been approved for early-stage disease and for BRCA mutations in the metastatic setting, it is worth considering how the efficacy of PARP inhibitors can be improved [51]. This study provides evidence that BUB1i increases the sensitivity of PARPi (olaparib) in BRCA mutant TNBC cells (Figure 6). This effect was compounded when combined with radiation (Figure 6), similar to what has been suggested by Kawanishi [52] and Feng [53] for BRCA mutant TNBC. Although recent studies have demonstrated that PARP inhibitor sensitivity does not depend on BRCA mutations [54], different cell lines have differential sensitivity/resistance to PARP inhibitors. Our findings demonstrate that BUB1i can sensitize different TNBC cell lines, irrespective of BRCA mutation status, to PARPi (Figure 3 and Figure 6). In the current study, we used a single PARP inhibitor, olaparib, which inhibits both PARP1 and PARP2. It would be interesting to evaluate the effect of BUB1i with additional PARP inhibitors that inhibit either PARP1 alone (talazoparib), PARP 1/2 both equally (niraparib), or PARP1/2/3 (rucaparib) [55]. Our findings are further supported by observations that CDK1 inhibitors [56] or androgen receptor inhibitors [57] increase PARPi sensitivity in breast cancers, while DNAPK inhibitors are effective at sensitizing TNBC to PARPi and IR [58].

PARP1/2 proteins usually detect SSB and recruit factors to repair the SSB. PARPi causes either PARP trapping on DNA break sites that leads to replication fork collapse and cell death (especially in BRCA mutant cell lines [59]) or PARPi can upregulate NHEJ and reduce HR, leading to genomic instability and cell killing [60]. Although the current studies were limited in their scope and we did not explore the mechanism of BUB1-mediated sensitization, we speculate that BUB1i sensitizes to PARP inhibitors because PARPi can increase the dependency on NHEJ, for which BUB1 is critical. Platinum agents (cisplatin, carboplatin) form DNA adducts that cause DNA replication errors leading from SSB to DSB, cell-cycle arrest in G1-S, and ultimately cell death [61,62]. Increased platinum-DNA adduct repair has been shown to be associated with cisplatin resistance [63]. Similarly, paclitaxel blocks the depolarization of microtubules, leading to improper chromosome segregation and G2/M cell cycle arrest [64,65], resulting in apoptotic cell death [66,67]. BUB1 not only regulates the cell cycle, but it also regulates DNA damage response through NHEJ. TP53 is mutated in the majority of TNBC [68]. BRCA deletion has been shown to cause changes in the level/mutations in p53 and BUB1 [69,70,71], suggesting that these may be regulated by the same mechanism; thus, BUB1 inhibition could sensitize TNBC tumors to chemotherapy, even in a p53 mutant background. Therefore, our findings in this study that BUB1 inhibition increases the cytotoxicity of paclitaxel, cisplatin, or olaparib with radiation in TNBC cell lines could be due to BUB1’s ability to target multiple pathways. Detailed mechanistic studies will be conducted in future to uncover exactly how BUB1 mediates chemoradiation sensitization. Our findings strongly support the nomination of BUB1 as a potential biomarker and a therapeutic target for chemoradiosensitization in TNBC.

## 5. Conclusions

The data presented here demonstrate that BUB1 inhibition sensitizes TNBC to a PARP inhibitor and radiation, irrespective of the BRCA mutation status. Moreover, inhibition of BUB1 synergistically sensitizes TNBC cell lines to cisplatin and paclitaxel with radiotherapy. Our studies nominate BUB1 as a novel molecular target for improving chemoradiation efficacy in TNBC.

## Figures and Tables

**Figure 1 biomolecules-14-00625-f001:**
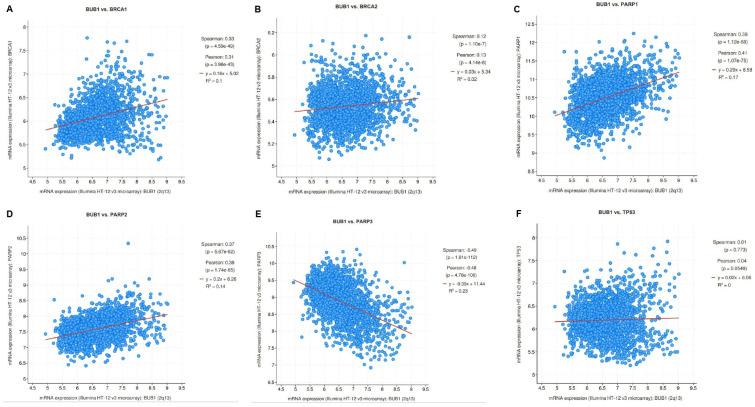
BUB1 expression demonstrate positive correlation with *BRCA1*, *BRCA2*, *PARP1*, and *PARP2* genes. mRNA expression plots showing correlation of BUB1 vs. (**A**) BRCA1, (**B**) BRCA2, (**C**) PARP1, (**D**) PARP2, (**E**) PARP3, and (**F**) TP53 in Breast cancer (METABRIC, 2509 samples) from cBioPortal.

**Figure 2 biomolecules-14-00625-f002:**
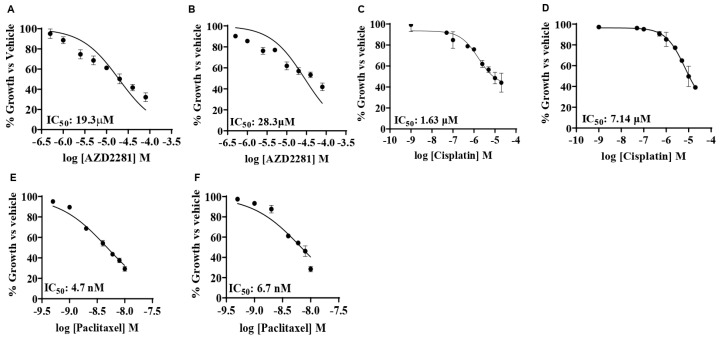
Cytotoxicity of anti-cancer agents on SUM159 and MDA-MB-231 cells by alamarBlueTM assay. IC50 of single agents in SUM159 (**A**,**C**,**E**) and MDA-MB-231 cells (**B**,**D**,**F**) were 19.3 μM and 28.3 μM (AZD2281/olaparib, panels **A**,**B**), 1.63 μM and 7.14 μM (cisplatin, panels **C**,**D**), 4.7 nM and 6.7 nM (paclitaxel, panels **E**,**F**).

**Figure 3 biomolecules-14-00625-f003:**
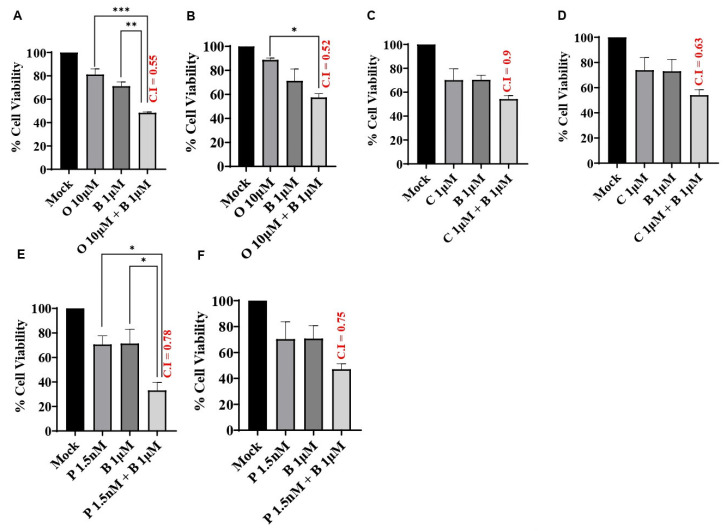
BUB1 kinase inhibitor synergistically sensitizes TNBC cells to olaparib, cisplatin, and paclitaxel. SUM159 (**A**,**C**,**E**) and MDA-MB-231 (**B**,**D**,**F**) cells were treated with AZD2281 (10 μM), Cisplatin (1 μM), and Paclitaxel (1.5 nM) in combination with BAY1816032 (1 μM). BUB1 inhibitor BAY1816032 showed synergistic effects (C.I. < 1) with all three classes of drugs. Difference between the groups was analyzed using 1-way ANOVA. P-values were defined as * *p* ≤ 0.05, ** *p* ≤ 0.01, *** *p* ≤ 0.001. Abbreviations: O—Olaparib/AZD2281, B—BAY1816032/BUB1 inhibitor, C—Cisplatin, P—Paclitaxel.

**Figure 4 biomolecules-14-00625-f004:**
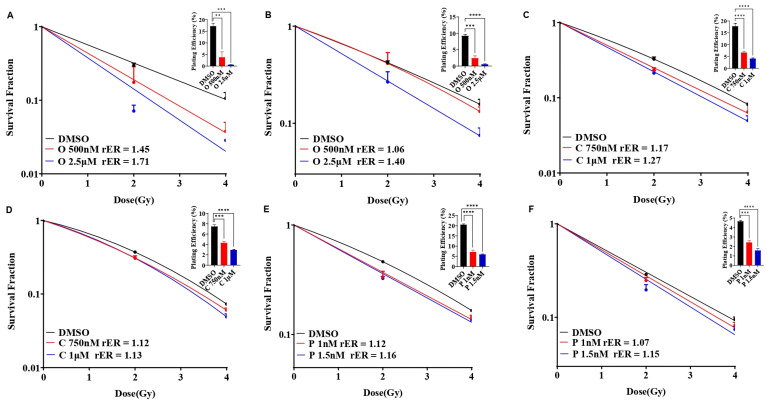
Olaparib, cisplatin, and paclitaxel radiosensitize SUM159 and MDA-MB-231 cells. SUM159 (**A**,**C**,**E**) and MDA-MB-231 (**B**,**D**,**F**) cells were treated with single agents AZD2281, Cisplatin, and Paclitaxel followed by ionizing radiation and clonogenic survival was estimated. Radiation enhancement radio (rER>1) indicates radiosensitization by single agent drugs. Plating efficiency graphs were plotted at 2 Gy. Difference between the groups was analyzed using 1-way ANOVA. P-values were defined as ** *p* ≤ 0.01, *** *p* ≤ 0.001, **** *p* ≤ 0.0001. Abbreviations: O—Olaparib/AZD2281, B—BAY1816032/BUB1 inhibitor, C—Cisplatin, P—Paclitaxel.

**Figure 5 biomolecules-14-00625-f005:**
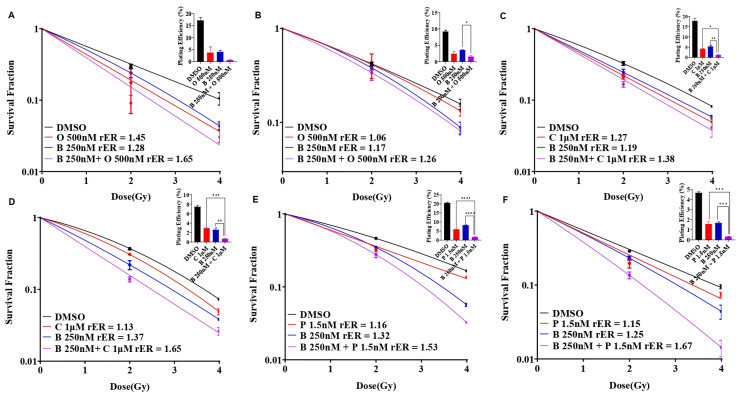
BAY1816032 enhances radiosensitization potential of olaparib, cisplatin, and paclitaxel in SUM159 and MDA-MB-231 cells. SUM159 (**A**,**C**,**E**) and MDA-MB-231 (**B**,**D**,**F**) cells were treated with BUB1 inhibitor BAY1816032 in combination with olaparib (A, B), cisplatin (**C**,**D**) or paclitaxel (**E**,**F**) followed by ionizing radiation. Cells were allowed to form colonies and radiation enhancement ratio (rER) was estimated. BUB1 inhibitor synergistically increases rER of olaparib, cisplatin, and paclitaxel (*p* = 0.99). Plating efficiency graphs were plotted at 2 Gy. Difference between the groups was analyzed using 1-way ANOVA. P-values were defined as * *p* ≤ 0.05, ** *p* ≤ 0.01, *** *p* ≤ 0.001, **** *p* ≤ 0.0001. Abbreviations: O—Olaparib/AZD2281, B—BAY1816032/BUB1 inhibitor, C—Cisplatin, P—Paclitaxel.

**Figure 6 biomolecules-14-00625-f006:**
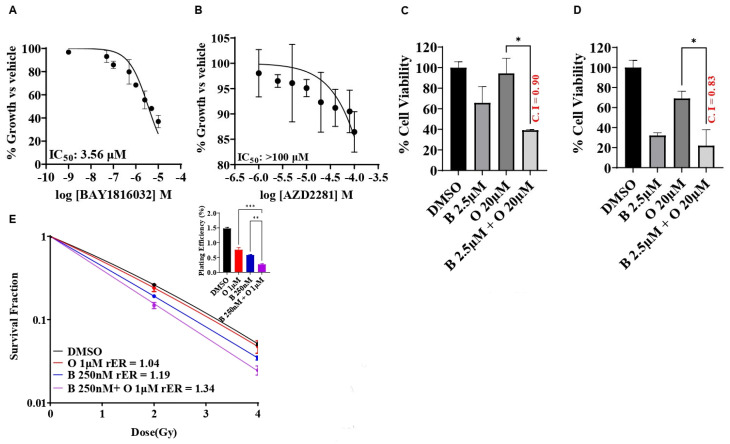
BAY1816032 sensitizes BRCA mutant TNBC cell line HCC1937 to PARP inhibitor and combination of these two inhibitors enhance radiosensitization in HCC1937 cells. Inhibitory effects of BAY1816032 (**A**) and Olaparib (**B**) at 72h on BRCA mutant HCC1937 cell line using alamarBlue assay. IC50 values of BAY1816032 and AZD2281 (Olaparib) in HCC1937 cell line were 3.56 μM and >100 μM, respectively. (**C**) BAY1816032 sensitizes HCC1937 cells to olaparib synergistically (CI < 1). (**D**) BUB1i mediated olaparib sensitization is enhanced with radiation (4 Gy). (**E**) HCC1937 cells were treated with BUB1 inhibitor and olaparib at lower concentration than IC50 ([BAY1816032]=250 nM, [AZD2281] = 1 μM) and irradiated. Cells were allowed to form colonies and radiation enhancement (rER) was estimated. BUB1i increased the radiosensitization by olaparib (*p* = 0.99). Plating efficiency graphs were plotted at 2 Gy. Difference between the groups was analyzed using 1-way ANOVA. P-values were defined as * *p* ≤ 0.05, ** *p* ≤ 0.01, *** *p* ≤ 0.001. Abbreviations: O—Olaparib/AZD2281, B—BAY1816032/BUB1 inhibitor.

## Data Availability

All the relevant data are already presented in the manuscript. Any additional data will be available upon request to the corresponding author.

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
