# Peer review of "BUB1 Inhibition Sensitizes TNBC Cell Lines to Chemotherapy and Radiotherapy"

_biomolecules, 2024, doi:10.3390/biom14060625_

Round 1

Reviewer 1 Report

Comments and Suggestions for Authors

This is a study conducted by Nyati et al. Author demonstrate that cell cycle checkpoint kinase BUB1 can serve as a novel molecular target for the treatment of triple negative breast cancer and form the basis for the development of rational combinatorial treatment approaches.

Several concern about this study:

1. Why does the author only focus on BUB1? What about BUB2?

2. Please add the purpose or aim of this study.

3. "We estimated the single agent IC50 of BUB1 inhibitor BAY1816032 to be less than 4 μM in these cells," and the author used 1 μM. The author should conduct a dose-response to determine the most effective dose. Additionally, author should also conduct experiments with different time points.

4. The author should add data for cell migration and proliferation.

5. This study lacks mechanistic insights.

Author Response

We thank the editor, editorial team and anonymous reviewers who provided helpful feedback on our manuscript. We have provided point by point response (in italics) to each of the critique below.

Reviewer #1: This is a study conducted by Nyati et al. Author demonstrates that cell cycle checkpoint kinase BUB1 can serve as a novel molecular target for the treatment of triple negative breast cancer and form the basis for the development of rational combinatorial treatment approaches.

We thank the reviewer #1 for their thorough evaluation of our manuscript and helpful suggestions that resulted in significant improvement.

  1. Why does the author only focus on BUB1? What about BUB2?

We thank reviewer #1 for this question. Initially, we performed a screen focused on the human kinome to identify kinases upregulated across 21 breast cancer cell lines that also impacted radiation sensitivity in human breast tumors (PMID: 25904749). We identified a list of 52 kinases whose expression was significantly elevated in triple-negative breast cancer (TNBC). To further characterize which of these 52 kinases played an important role in the aggressive features of TNBC, we combined expression, phenotypic, and clinical outcomes data to prioritize kinases that warranted further interrogation.  We prioritized those kinases that had the highest level of differential expression in TNBC, showed limited to no expression in normal tissues (including the mammary gland, thus were specific for breast cancer), were associated with clinically relevant outcomes, and for which we would be able to obtain or generate a specific inhibitor that was of clinical-grade quality to aid in translational efforts. To that end, BUB1 was one of the top 4 nominated kinases as it showed significantly elevated expression in triple-negative and basal-like breast cancers and limited expression in normal tissues (Fig. 1C-D, doi: https://doi.org/10.1101/2024.05.07.592812)

An inhibitor against BUB1 with clean safety profile is available. We have been actively studying numerous roles for BUB1 in carcinogenesis for several years and have demonstrated that BUB1 inhibitor is well tolerated and radiosensitizes TNBC xenografts.

BUB2 is a yeast cell cycle checkpoint protein (not a mammalian protein) and does not have a specific inhibitor developed yet. Additionally, certain studies indicate that BUB2 may have a distinct function in mitotic checkpoint activation compared to BUB1 protein (PMID: 10352016). We did not work on BUB2 since it is not a mammalian protein, no role has been identified in human cancers and there is no specific inhibitor available.

  1. Please add the purpose or aim of this study.

We apologies that the purpose of our study was unclear to the reviewer. We have now included a section that clearly mentions rationale and objectives of our study. Please refer Page 4, line 97-102. The main objective of this study was to ascertain if BUB1 inhibition improved the effectiveness of chemotherapy and chemoradiation in TNBC models. Given BUB1’s strong correlation to aggressiveness and different classes of drugs (PMID: 38396175), and our observation that BUB1 inhibition sensitizes TNBC to radiation and lung cancers to chemoradiation (doi: https://doi.org/10.1101/2024.05.07.592812, PMID: 38712071), we rationalized that combining BUB1 inhibitor would provide strong chemoradiation sensitization in TNBC. Indeed, our results from this study confirmed that BUB1 inhibitor sensitizes TNBC to different classes of drugs (platinum, PARPi, microtubule depolarization inhibitor). This research provides compelling evidence for nominating BUB1 as a novel molecular target to enhance the effectiveness of chemoradiation in TNBC.

  1. "We estimated the single agent IC50 of BUB1 inhibitor BAY1816032 to be less than 4 μM in these cells," and the author used 1 μM. The author should conduct a dose-response to determine the most effective dose. Additionally, author should also conduct experiments with different time points.

Recently we performed dose finding study (low nano-molar to high micro molar concentrations of BUB1i BAY1816032) in several TNBC cell lines (https://doi.org/10.1101/2024.05.07.592812), (Figure 2A-F) and identified IC50 ranging from 1.6 μM to 3.9 μM. We estimated that in SUM159 and MDA-MB-231 cell lines, the IC70 value was around 1 μM. Usually drug concentrations at or near IC70 are used in drug-combination studies. This is why in all our combination studies with olaparib, cisplatin, and paclitaxel, we used 1 μM BUB1i (Figure 3).

We perform cell proliferation assays by treating cells with BUB1i for 72 h, and we also performed cell proliferation assays with BUB1i for 7 days in cell lines that don’t form proper colonies such as small cell lung carcinoma (SCLC) cell lines PMID: 38712071 (Figure 2F-G). For clonogenic assays cells are treated with BUB1i 1h prior to radiation, and cells/colonies are fixed after 7-23 days depending on the cell line. Some cell lines form well-separated colonies within days while other takes up to 3 weeks. Thus, we have performed experiments with various time points using BUB1i.

We perform cell-survival assays (in 96-well plates) to identify the IC50 values and use less drug concentrations in clonogenic cell survival assays with or without radiotherapy. The cells are hypersensitive to drugs when seeded at very low densities for colony formation and when radiation is applied. The use of significantly reduced drug concentrations in colony formation assays (clonogenics; gold standard in radiation biology) demonstrates that one needs significantly lower drug concentrations (sometimes up to 2 logs lower than IC50) to kill cancer cells in these assays. This is why there is difference in drug concentrations between cell proliferation and clonogenic survival assays.

  1. The author should add data for cell migration and proliferation.

Cell proliferation studies were performed in various TNBC cell lines using BUB1i in our recent study. Please refer Figure 2A-F in this preprint (doi: https://doi.org/10.1101/2024.05.07.592812). In our previous studies we demonstrated a role for BUB1 in mediating cell migration and cell proliferation (PMID: 25564677). The cell migration (8h time point) and invasion assays (24h time point) were performed after knocking down BUB1 by siRNA (PMID: 25564677). Depletion of BUB1 inhibited both migration and invasion (please see Supplementary Figure S6 in this article). Other investigators have also reported reduced cell migration after BUB1 inhibition (PMID: 34337852).

  1. This study lacks mechanistic insights.

The main purpose of this study was to determine whether BUB1i improved the efficacy of chemotherapies and targeted agents (Olaparib, Cisplatin, and Paclitaxel) synergistically in TNBC models. Knowing that it does, we are now in the process of generating mechanistic data to secure extramural (NIH) funding. Given BAY1816032's strong tolerability and desirable pharmacokinetic profile, we hope to translate the in-vitro (cell based) drug combination findings to relevant in-vivo tumor models in future along with mechanistic studies.

Reviewer #2: This manuscript presents in vitro inhibition of TNBC cell survival following combinations of a BUB1 inhibitor with radiation and either olaparib, paclitaxel, or cisplatin. These studies aimed to determine the effects of BUB1 inhibition on the sensitivity of TNBC cells to established therapies. Adequate discussion of relevant literature is presented. The following items can be considered by the authors to address several questions raised during review of the manuscript.

We thank the reviewer #2 for their overall positive feedback on our work.

  1. Please correct the x-axis values/labels for Figure 2 (panels A, B, E, and F).

We thank the reviewer for pointing it out. We have re-created the plots and corrected the x-axis values in Figure 2 to be consistent.

  1. It seems that ANOVA would be more appropriate for statistical analyses rather than the reported student’s t-test (since in many cases there were statistical comparisons for 3 conditions when determining the combination index).

We have now performed statistical analysis using 1-way ANOVA as recommended by the reviewer. The statistical significance values are provided in figures as well. We have also updated the Material and Methods, Results, and Figure Legends where appropriate. Please refer Page 7, line 151-152, 166-167, Page 9, line 202, and legends of Figure 3-6.

  1. Please provide results for determining the IC50 (A+B) when using drugs in combination, since this value was used to determine the combination index; the data provided show cell growth inhibition of the combined agents, but the data do not correspond to determinations of the IC50 values when the drugs were combined since not every combination resulted in 50% cell growth inhibition at the drug doses presented.

We thank the reviewer for pointing out our error. Earlier we wrote C.I. index formula incorrectly under "Material and Methods" section. We have now corrected it in the revised manuscript and accordingly also updated the “Results” section too. As mentioned above for our response to reviewer #1, point #3, IC70 concentrations were used for drug-combination experiments (and not the IC50). Please refer Page 6, line 146-150 and Page 9, line 206. Furthermore, Figure 3 incorrectly labels Olaparib as 1 μM while it should be as 10 μM. We have corrected it now in revised manuscript.

  1. Please comment on results where it appears that there were non-significant differences in cell growth inhibition between BUB1 inhibitor alone and combination of BUB1 inhibitor with chemotherapy (Fig. 3D, Fig. 6C, D).

As suggested by the reviewer, we have now included additional information on these figures under ‘Results’. Please refer Page 9, line 201-204, Page 10, line 231-232, and Page 11, line 256.

Additional minor comments:

-Fig.3, 4, 5, 6: please add suitable abbreviation definitions (P, O, B, etc.) to the figure captions to indicate each drug, and please add spacing in the figures between the abbreviation and the concentration (revise “O1 uM” to “O 1uM”, etc.) 

Suitable abbreviations are now included in the figure captions such that they are easy to read and interpret.

-Please verify complete citation information is present and correct for all references (some appear to be missing article numbers, such as references 2, 6, 25, etc. And references 34 and 42 are duplicates).

We've updated and corrected the references as the reviewer advised.

-The language has adequate quality to be readily understood; a few minor grammatical errors are present, but these do not detract significantly from the overall content.

A native English speaker has proofread the entire manuscript and corrected grammatical errors.

Reviewer 2 Report

Comments and Suggestions for Authors

This manuscript presents in vitro inhibition of TNBC cell survival following combinations of a BUB1 inhibitor with radiation and either olaparib, paclitaxel, or cisplatin. These studies aimed to determine the effects of BUB1 inhibition on the sensitivity of TNBC cells to established therapies. Adequate discussion of relevant literature is presented. The following items can be considered by the authors to address several questions raised during review of the manuscript.

Specific comments to be addressed:

1.    Please correct the x-axis values/labels for Figure 2 (panels A, B, E, F).

2.    It seems that ANOVA would be more appropriate for statistical analyses rather than the reported student’s t-test (since in many cases there were statistical comparisons for 3 conditions when determining the combination index).

3.    Please provide results for determining the IC50(A+B) when using drugs in combination, since this value was used to determine the combination index; the data provided show cell growth inhibition of the combined agents, but the data do not correspond to determinations of the IC50 values when the drugs were combined since not every combination resulted in 50% cell growth inhibition at the drug doses presented.

4.    Please comment on results where it appears that there were non-significant differences in cell growth inhibition between BUB1 inhibitor alone and combination of BUB1 inhibitor with chemotherapy (Fig.3D, Fig.6C,D).

Additional minor comments:

-Fig.3, 4, 5, 6: please add suitable abbreviation definitions (P, O, B, etc.) to the figure captions to indicate each drug, and also please add spacing in the figures between the abbreviation and the concentration (revise “O1 uM” to “O 1uM”, etc.) 

-Please verify complete citation information is present and correct for all references (some appear to be missing article numbers, such as references 2, 6, 25, etc. And references 34 and 42 are duplicates)

Comments on the Quality of English Language

The language has adequate quality to be readily understood; a few minor grammatical errors are present, but these do not detract significantly from the overall content.

Author Response

(The authors gave the same response as above.)
